# Use of the Therapy App Prescinde for Increasing Adherence to Smoking Cessation Treatment

**DOI:** 10.3390/healthcare11243121

**Published:** 2023-12-08

**Authors:** Francisca López-Torrecillas, Isabel Ramírez-Uclés, María del Mar Rueda, Beatriz Cobo-Rodríguez, Luis Castro-Martín, Sabina Arantxa Urrea-Castaño, Lucas Muñoz-López

**Affiliations:** 1Center for Mind, Brain, and Behavior Research (CIMCYC), University of Granada, 18071 Granada, Spain; fcalopez@ugr.es; 2Department of Personality, Assessment and Psychological Treatment, National Distance Education University (UNED), 28040 Madrid, Spain; 3Department of Statistics and Operations Research, University of Granada, 18071 Granada, Spain; mrueda@ugr.es (M.d.M.R.); sarantxauc@correo.ugr.es (S.A.U.-C.); 4Department of Quantitative Methods for Economics and Business, University of Granada, 18071 Granada, Spain; beacr@ugr.es; 5Health and Consumption Counseling, Andalusian School of Public Health, 18011 Granada, Spain; luiscastro193@ugr.es; 6Department of Personality, Assessment and Psychological Treatment, University of Granada, 18071 Granada, Spain

**Keywords:** mobile app, tobacco, cannabis, physical exercise, adherence

## Abstract

Tobacco use poses major health risks and is a major contributor to causes of death worldwide. Mobile phone-based cessation apps for this substance are gaining popularity, often used as a component of traditional interventions. This study aimed to analyze adherence to an intervention using a mobile phone application (App-therapy Prescinde (v1)) as a function of sociodemographic variables (age, gender, educational level, and profession) as well as the primary activities supported by the app (reducing tobacco or cannabis use and increasing physical exercise). The participants were recruited through the web pages of the Occupational Risk Prevention Service and the Psychology Clinic of the University of Granada during the COVID-19 confinement period. The application’s contents include three components (self-report, motivational phrases, and goal setting). Our findings indicate that being male, being aged between 26 and 62, having a high school education, and being unemployed increase the likelihood of adherence to the Prescinde therapy app three months after usage. Our findings highlight the importance of developing new therapeutic approaches and conducting in-depth studies on the factors associated with adherence to tobacco cessation and cannabis cessation treatments via mobile phone applications.

## 1. Introduction

Tobacco use poses significant health risks and is a major contributor to causes of death worldwide. Every year, approximately nine million people die as a result of tobacco use, which represents around half of all tobacco users [1,2]. The International Agency for Research on Cancer [3,4] has identified 117 carcinogenic substances: 37 in unburned tobacco and 80 in tobacco smoke. Despite the significant risks posed by tobacco to both individuals and the environment [5], no notable decline in tobacco use has been observed to date [6]. Urgent measures are therefore required to combat smoking-related diseases, including cancer, chronic bronchitis, respiratory problems, coronary heart disease, and cardiovascular conditions. Furthermore, it is crucial to address the environmental damage caused by smoking. Additionally, tobacco and cannabis rank among the most widely used substances among young individuals aged 14 to 18 years. The early onset of the use of these substances increases rapidly during adolescence [7] and has been associated with a wide range of adverse mental health-related effects such as psychosis, mood disorders, and impaired cognitive function [8,9].

Cognitive behavioral treatment is widely recognized as one of the most effective therapeutic approaches for individuals seeking to quit using these substances [10,11,12]. However, there is a concerning decline in the number of people accessing face-to-face support for these addictions [13]. Only one in six people struggling with addiction to these substances seek cessation treatment. The most common method for smoking cessation is without assistance, as in, for example, self-guided nicotine patch use and/or medications for nicotine replacement without prescription [14,15]. Unfortunately, this self-guided approach is often accompanied by the excessive use of over-the-counter drugs [15,16].

The treatment for quitting cannabis and/or tobacco use continues to pose challenges (cognitive behavioral therapy, nicotine replacement therapy, or nicotine patches), as indicated by low treatment adherence rates ranging from 13% to 50% [10,17,18], while high treatment relapse rates ranging from 40% to 60%, even as high as 86% [19,20], seem to further compound the issue. Only one in six people with addiction to these substances seek treatment to stop using them. In addition, both treatment dropouts and relapses contribute to repeated treatment demands or new re-admissions. This places a strain on health systems, leading to increased costs and hampering the management of treatment centers while potentially limiting the health care provided to patients [21].

For these reasons, there is a pressing need to develop new therapeutic approaches and to study in depth the factors that influence adherence to tobacco and cannabis cessation treatments. Recent studies [5,14,18,22,23,24,25,26] have shed light on the relationship between attempts to quit these substances and treatment adherence, highlighting the influence of sociodemographic characteristics such as socioeconomic status, age, and gender. Specifically, age has been found to be directly linked to treatment adherence [14], and women face more difficulties when attempting to quit smoking and maintain adherence to treatment [27]. Additionally, educational level and income have been identified as indicators of smoking risk [28]. Moreover, a lower socioeconomic level and working conditions involving night shifts and low pay have shown an inverse relationship with treatment adherence [29]. On the other hand, physical exercise emerges as a positive factor in smoking cessation. Engaging in physical exercise has been found to reduce cravings during the process of quitting smoking. Even a single session of physical exercise can significantly reduce the desire to smoke and is also associated with a lower likelihood of relapse [30,31].

The widespread availability of internet access and the rising prevalence of smartphone use have paved the way for digital interventions, including cell phone apps, which offer cost-effective treatment delivery, improved intervention strategies, and optimized access to evidence-based information on treatments aimed at cannabis and/or tobacco smoking cessation [32,33]. The recent use of these applications in the health care sector has increased participant engagement, leading to significant behavioral changes such as increased physical exercise [34], weight loss [35], and the successful cessation of tobacco and cannabis use [36,37]. Cell phone-based smoking cessation interventions provide several advantages, including unrestricted access to specialized information and overcoming geographic barriers or usual programming restrictions. These interventions offer privacy and convenience and are cost-effective [38]. Furthermore, they have the potential to reach large audiences and expand access to smoking cessation treatment [39]. Smoking cessation apps offer a wide range of content options, ranging from videos and short and simple ads to more comprehensive programs encompassing mindfulness meditation, self-efficacy enhancement, motivation for change, and smoking cessation counseling (through educational, self-efficacy, and coping skills messages). Additional components include messages promoting happiness based on positive psychology principles, aimed at increasing adherence to treatment [13,26,29,39,40,41]. These apps can also serve as adjuncts to cognitive behavioral treatments [42]. However, a content analysis on smoking cessation apps has revealed several shortcomings and little evidence-based research [43]. Thus, recent published studies related to smoking cessation apps still fail to analyze factors such as user interaction, design elements, and their impact on engagement with these apps. Moreover, there is a need to investigate the user profiles that are most likely to engage with these apps.

This study aimed to analyze adherence to the intervention using a cell phone application called App-therapy Prescinde. The analysis considered various sociodemographic variables, including age, gender, educational level, and profession, as well as the main activities that users engaged in, specifically a reduction in tobacco or cannabis consumption and an increase in physical exercise. The specific objectives were (1) to analyze the users’ continued engagement with the application according to the main activity performed (reduction in tobacco or cannabis consumption and increase in physical exercise); (2) to analyze the dropout rate during the study period; (3) to analyze the frequency of app use according to sociodemographic variables, the main activity performed in the application, and the duration of engagement; (4) to study the likelihood of users remaining engaged with the app over the study period according to sociodemographic variables and main activity pursued; and (5) to examine adherence levels at one and three months according to sociodemographic variables and the main activity performed on the app.

## 2. Materials and Methods

### 2.1. Design

Respondent-driven sampling [44,45] was employed to select the initial sample of participants. This sampling method targeted individuals who were willing to participate in the study and were representative of the population of tobacco and/or cannabis users who wanted to stop their consumption, as well as those who aimed to increase their frequency of physical exercise. The initial sample was provided with a link to the application, which they shared with other individuals in similar circumstances, forming subsequent waves of recruitment. The seed sample was recruited through the web pages of the Occupational Risk Prevention Service and the Psychology Clinic of the University of Granada during the period of confinement and state of emergency (20 March 2020 until 10 January 2022). This study was approved by the Ethics Committee of the University of Granada. The sample selection procedure was not probabilistic. In recent years, non-probability surveys have had great growth, but a problem arises when selecting the sample, and it is the lack of an adequate sampling frame which causes us to have selection bias if the population covered differs from the target population. Given this problem, there are statistical procedures to adjust these biases, such as calibration and propensity score adjustment [46,47,48], and thus enable us to generalize the results of the sample to the population under study.

### 2.2. Data Collection

The specific data collection process was as follows: Initially, all participants accessed a link and downloaded the app from Telegram (https://web.telegram.org/). The participants then completed a form with information regarding their age, gender, educational level, and occupation (if currently studying and/or working and their profession). Subsequently, participants had to choose to access the Prescinde therapy app to (1) stop using cannabis, (2) stop smoking tobacco, or (3) perform physical exercise. Depending on the option chosen, the participants answered questions related to their cannabis and/or tobacco use and physical exercise, were encouraged to create self-recordings, and received motivational phrases to support their progress.

### 2.3. Description of the Therapy App Prescinde

This application is directly accessible through the link https://prescinde.github.io/ (accessed on 1 November 2023) for easy distribution and use on any device or operating system (such as Android, iOS, Windows or Linux). We developed the entire Prescinde therapy app using the JavaScript programming language. In particular, the server is implemented in Node.js and the database in MongoDB.

When accessing the application, the welcome screen appears so that the user can link the therapy app Prescinde to their Google account. This process facilitates the saving of registration information, making it easy to transfer data when switching devices. Additionally, the application is linked to Telegram (instant messaging platform). Figure 1 shows the initial screen of the app.

The application includes three components. The first component involves self-reporting the main activity. The second component concerns the presentation of motivational phrases, which are tailored according to the main activity reported by the user and can vary depending on whether this activity is increasing or decreasing. The third component focuses on presenting goals and inquiring about the fulfillment of these goals.

During the self-recording of the main activity chosen (stopping cannabis and/or tobacco use, or physical exercise), the participants were required to provide specific details. For cannabis and tobacco use, the participants had to indicate when and where they smoked and rate their desire on a scale from 1 to 10, with very 1 being low and 10 being very high. In the case of physical exercise, the participants had to record the exercise performed and their degree of satisfaction. Figure 2 shows a screenshot of a question on physical exercise and Figure 3 shows a screenshot of the self-reporting interface for tobacco and cannabis consumption.

Regarding the second component, which involves the presentation of motivational phrases, these depend on the chosen main activity chosen and whether there is an increase or decrease in this activity. In the case of cannabis consumption, if the user did not reduce their consumption, several phrases were displayed. These phrases highlighted the negative effects of cannabis consumption, such as reduced concentration, impaired memory and cognitive performance, increased fatigue, distorted perception of time and space, depressive effects, increased appetite, psychomotor difficulties, social isolation, heightened risk of anxiety, panic, paranoia, psychosis, schizophrenia, and even suicide attempts. Furthermore, these phrases emphasized the increased risk of bradycardia (abnormally slow heart rate below 60 beats per minute), a lack of motivation, feelings of self-disgust, embarrassment, disappointment, and feelings of tension and/or concern related to relationship problems (family, partner, or friendship) caused by cannabis consumption. On the other hand, if the user reduced their consumption, motivational phrases were displayed to encourage and congratulate them. These phrases included the following: “You can do this! You are capable of reducing the consumption of joints if you think about it firmly enough. You do not need to use to feel good. Congratulations, you are a champion. From now on you will be able to see how your self-esteem improves, along with your social interaction skills; your ability to concentrate; your ability to think critically; your self-control; and your learning and memory. You can express your opinions and decisions in a more assertive way (greater self-control over mood and behavior). In addition, quitting smoking joints reduces the risk of psychosis and experiencing altered perceptions of reality, and reduces the risk of suffering anxiety, panic, or depression”.

When the user chose to quit smoking as their main activity, the motivating phrases in the event of not reducing consumption were as follows: “Tobacco consumption increases the risk of cardiovascular diseases and various types of cancer. It can lead to impotence, reduced fertility, osteoporosis, cataracts, hearing loss, and gastric ulcer, among other health issues. Smoking causes bad breath that makes others uncomfortable, leading to social rejection. It also increases the appearance of wrinkles. Your surroundings are affected by second-hand smoke. Smoking slows you down and drains your financial resources. The tobacco industry profits at your expense. Are you willing to sacrifice your life for it?” On the other hand, the motivational phrases for the case where the user reduces their tobacco consumption are as follows: “Congratulations, you are a champion. From now on you will be able to see how your lung capacity, blood circulation, skin hydration, taste and smell improve. By reducing tobacco use, your risk of heart attacks, premature wrinkles, cancer, and other health risks decreases. You will gain social acceptance, increased energy, improved physical capacity, and overall enhancements to your physical and mental well-being. Moreover, you will reclaim you independence and no one will breathe your noxious smoke”.

If the user chose the main activity of increasing physical exercise, the motivating phrases were as follows: “Physical exercise improves your self-esteem and self-control, your physical fitness, insulin resistance, flexibility, and joint mobility, while helping you to maintain a healthy body weight. It regulates blood pressure, increases or maintains bone density, increases muscle tone and strength, and enhances your concentration, memory, cognitive performance, and emotional well-being. Moreover, exercise reduces anxiety, stress, depression, and the risk of suffering from panic attacks”.

The third component of the Prescinde therapy app focuses on presenting goals related to seven areas: (1) improving relationships with family; (2) improving relationships with friends; (3) health; (4) leisure; (5) study; (6) job search; and (7) work. To assess the achievement of these goals in each of the seven areas, a set of questions is included. Users are required to indicate whether or not they have completed specific tasks/activities. For area (1), *improving relationships with family*, the following questions were included: Have you responded politely? Have you avoided bringing up past conflicts? Have you respected decisions, opinions, and ideologies different from your own? Have you planned activities to address any discomfort transmitted by your family members? The questions related to area (2), *improving relationships with friends,* included the following: Have you responded in a polite way? Have you avoided bringing up past conflicts? Have you respected decisions, opinions, and ideologies different from your own? Have you planned activities to address any discomfort transmitted by friends? The questions related to area (3), *health*, were as follows: Have you meditated or relaxed? Have you addressed your concerns? Have you walked to different places? Have you cycled to different places? Have you engaged in relaxation exercises? Have you drunk 2 L of water per day? Have you eaten five light meals a day (breakfast, mid-morning snack, lunch, afternoon snack, and dinner)? Have you eaten fruits and vegetables (5 portions a day)? Have you eaten foods rich in fiber (whole grain products)? Have you played any outdoor sports or gone to the gym? Have you slept for 8 h? The questions related to area (4), *leisure,* were: Have you gone to the movies? Have you gone out for tapas, lunch, dinner, or coffee or similar? Have you visited the beach, the countryside, or similar? Have you enrolled in any course or activity in photography, radio, music, or painting? Have you actively participated in any environmental movement or similar? Have you done your household chores (washing dishes, making the bed, preparing food, among others)? The questions related to area (5), *study,* were as follows: Have you attended class every day? Have you studied for at least three hours a day? Have you completed the practical tasks and activities in groups or individually, depending on the suggestion of the teacher? Have you practiced the class exercises at home? Do you follow a pre-determined study timetable? The questions related to area (6), *job search,* were the following: Have you gone to places to leave your curriculum vitae of your professional profile? Have you gone to careers guidance? Have you registered your details on job sites on the Internet as well as on specific employer pages? Have you completed face-to-face or online courses focused on your professional profile? Have you looked at job offers from private companies? Have you registered in the SAE? Have you looked for a job in the SAE? Have you visited the *Andalucía Orienta* website or similar? Have you visited the *empléate* website or similar? Have you visited the portal of the Junta de Andalucía or another community and clicked on the job opportunities tab? The questions related to area (7), *work,* were as follows: “Have you made a to-do list? Have you planned your activities? Have you identified your stress triggers or events that generate negative emotions and developed a healthy response? Have you established boundaries between your work and personal life? Have you avoided arguing with your colleagues? Have you effectively managed interactions with your boss?” These questions were distributed evenly throughout the week. The application incorporated integrated forms to track the personalized progress of each user. A screenshot of the physical exercise history is shown in Figure 4.

### 2.4. Participants

This study consisted of 166 participants who used the application from 20 March 2020 to 10 January 2022. Of the sample, 80.72% were women and 19.38% were men. The participants were divided into three groups according to their age, which ranged from 17 to 62 years. Group 1 (comprising 57.22% of the participants) was aged between 17 and 18 years, Group 2 (21.69% of the participants) was aged between 19 and 25 years, and Group 3 (21.08% of the participants) was aged between 26 and 62 years. The main activities for which the participants used the Prescinde therapy app were physical exercise (68.07%), quitting smoking tobacco (28.31%), and quitting smoking cannabis (3.61%). Most of the participants were students (87%) and women (73%). The participants’ main level of education was high school (49.4%), bachelor degree (37.3%), and master and doctorate (13.3%). Table 1 provides sociodemographic information as well as details of app usage, including time spent and adherence.

### 2.5. Measurements and Variables

#### 2.5.1. Study Variables

-Frequency of use (number of times the participant uses the app);-Time spent on the app (time), divided into 4 categories: (1) 1 week, (2) 1 month, (3) 3 months, and (4) more than 3 months;-Adherence1, understood as having been using the app for at least 1 month, categorized as 1 (adherence) and 0 (non-adherence);-Adherence3, understood as having been using the app for more than 3 months, categorized as 1 (adherence) and 0 (non-adherence).

#### 2.5.2. Sociodemographic Variables

-Gender, divided into two groups: (1) male, and (2) female;-Academic status (studying), divided into 2 categories: (1) no or (2) yes;-Employment status (working), divided into 2 categories: (1) no or (2) yes;-Tobacco use, divided into 2 categories: (1) no or (2) yes;-Cannabis use, divided into 2 categories: (1) no or (2) yes;-Age, divided into 3 groups: Group 1 (aged between 17 and 18 years), Group 2 (aged between 19 and 25 years), and Group 3 (aged between 26 and 62 years);-Main activity on the app, divided into 3 categories: (1) stop smoking tobacco (tobacco), (2) stop smoking cannabis (cannabis), and (3) physical exercise (physical exercise);-Level of education, divided into 3 categories: (1) high school; (2) bachelor degree, and (3) master/doctorate;-Occupation, divided into 8: (1) public administration, (2) agriculture/fishing, (3) trade, (4) private company manager, (5) hospitality/tourism, (6) technical professional, (7) transportation, and (8) unemployed.

## 3. Results

### 3.1. Time of Adherence

Figure 5 shows the number of participants who continued using the application according to the activity they engaged in (reducing tobacco or cannabis consumption or increasing physical exercise). Our findings indicate that 104 (62.65%) participants remained on the Prescinde therapy app for one week, 60 (36.14%) for one month, and 22 (13.25%) for three months or more.

### 3.2. Frequency of Use

Table 2 provides an overview of usage frequency according to gender, age, and main activity (reducing tobacco and/or cannabis consumption and increasing physical exercise) and the time spent on the app (one week, one month, three months, or more than three months). It was observed that 21% of participants used the app only once, while the maximum usage by a participant was 266 times (0.6%). The mean number of app uses was 28.77, and the median was seven. Regarding gender differences, men utilized the app more frequently across the entire analysis period (1 week, 1 month, 3 months, or more than 3 months). In terms of age, Group 3 (aged 26 to 62 years) showed higher app usage than Group 2 (aged 19 to 25 years), while Group 1 (aged 17 to 18 years) displayed the lowest usage.

During the first week, the most commonly used activity with the app was to stop using cannabis, followed by quitting smoking. After one month, the participants primarily used the app to quit smoking tobacco, followed by increasing their levels of physical exercise. At the three-month mark, participants aiming to quit smoking tobacco used the app more frequently than those wanting to quit cannabis use and those focusing on increasing their physical exercise. In addition, over a time period of more than three months, engaging in increased physical exercise was the most frequent use of the app, followed by quitting smoking.

### 3.3. Probability of Total Adherence as a Function of Gender and Age: Survival Analysis

We performed a survival analysis to study the probability of remaining in the application over time as a function of gender and age. The survival functions were calculated using the Kaplan–Meier method [49]. In Figure 6, we can observe that the likelihood of the participants adhering to use of the app begins to stabilize at around 70 days.

To compare the survival curves according to the gender of the participants, we conducted the Log-Rank test [50]. Figure 7 shows the survival functions according to gender. Although no statistically significant differences were found (*p* = 0.200), it is worth noting that during the first 11 days, the probability of adherence among men was lower. However, from day 11, this probability shifted, with women showing the lowest probability of adherence.

Significant differences in survival functions were found according to age (Log-Rank test confirmed a significant difference between age groups (17–18; 19–25; 26–62) (*p* = 0.009)). Figure 8 illustrates that within the first 17 days, Group 3 (aged 26 to 62) showed the lowest probability of continued app usage, followed by Group 1 (aged 17 to 18). Conversely, Group 2 (aged 19 to 25) had the highest probability of continued app usage. However, as the duration of app usage increased, a direct relationship between age and the probability of continued app usage emerged, particularly after day 21.

### 3.4. Probability of Adherence Based on the Main Activity: Survival Analysis

The Log-Rank test revealed statistically significant differences (*p* = 0.002) in survival functions among the groups categorized according to their main app activity (quitting cannabis/smoking or increasing physical exercise). Figure 9 shows that participants who used the Prescinde therapy app to quit smoking cannabis showed a decrease in the probability of continued app usage until day 41, while those who used the app to quit smoking tobacco remained active until day 142. Conversely, participants who used the app to increase physical exercise showed a higher probability of continued usage, with their survival function extending until day 731. Please note that Figure 9 has been limited to approximately 100 days to provide a clear visualization of the differences in survival curves.

### 3.5. Nonparametric Analysis of Continuous Variables: Frequency of Usage and Time Elapsed from the First to the Last Usage of the Prescinde Therapy App and the Relationship with Sociodemographic Variables

A non-parametric analysis was conducted to examine the relationship between quantitative variables, such as frequency of use, and time elapsed from the first to the last usage of the therapy app Prescinde with sociodemographic variables. Since the data did not follow a normal distribution [51,52], the Mann–Whitney–Wilcoxon test [53,54] was employed for the sociodemographic variables with two categories, namely gender, educational and/or employment status, and tobacco and cannabis use. For variables with more than two categories, including age, main activity when using the app, educational level, and profession, the Kruskal–Wallis test was used [55]. The results can be seen in Table 3.

These analyses revealed statistically significant differences in the frequency of use of the app for quitting tobacco (*p* = 0.015) and cannabis (*p* = 0.012), according to age (*p* = 0.048) and education level (*p* = 0.017). Furthermore, concerning the time variable, significant differences were also found according to education level (*p* = 0.021).

### 3.6. Nonparametric Analysis of Qualitative Variables: Adherence and Its Relationship with Sociodemographic Variables

A Chi-square test was employed to examine adherence and its relationship with sociodemographic variables including gender, educational and/or employment status, tobacco and cannabis use, age, main activity when using the application, educational level, and profession. This test was chosen as the data did not follow a normal distribution [56] and the results can be seen in Table 4.

Statistically significant differences were observed in adherence for more than 3 months according to the variables of tobacco use (*p* = 0.010), cannabis use (*p* = 0.010), age (*p* = 0.010), main activity for which the app is used (*p* = 0.012), educational level (*p* = 0.010), and profession (*p* = 0.034).

Post hoc tests were used to identify significant pairwise differences between the variables analyzed (frequency of use, elapsed time and adherence (more than 1 month and more than 3 months), and their relationship with the sociodemographic and activity variables of the Prescinde therapy app (age, main activity, educational level, and profession, main activity, educational level and profession)). The Mann–Whitney-Wilcoxon test was employed to analyze quantitative variables (frequency of use, elapsed time), while the Chi-square test was used for qualitative variables (adherence for more than 1 month and more than 3 months). For the Mann–Whitney–Wilcoxon test, the Bonferroni, Holm, Hommel, and Hochberg corrections were applied. However, based on the similarity of the results obtained (compared with the other tests), the Bonferroni correction was selected for simplicity in interpreting the findings. Table 5 shows the results of the Hommel correction, specifically for significant comparisons. Notably, a relationship was found between remaining in treatment for more than 3 months (Adherence3) and age. Specifically, Group 3 (aged 26 to 62) had a higher likelihood of remaining in treatment than Group 1 (aged 17 to 18).

With regard to the relationship between remaining in treatment for more than 3 months (Adherence3) and the main activity on the app (activity carried out to quit cannabis/tobacco or to increase physical exercise), the contrasts showed statistically significant differences between the participants who had chosen to stop using tobacco as their main activity and the participants who had chosen to stop using cannabis as their main activity, with the participants who had chosen to stop using tobacco having the highest probability of remaining in treatment for more than 3 months. Thus, the contrasts also showed statistically significant differences between the participants who had chosen to stop using tobacco as their main activity and those who had chosen physical exercise as their main activity, with the participants who had chosen physical exercise having the highest probability of remaining in the study. No statistically significant differences were found for the contrasts between the participants who had chosen to stop using cannabis as their main activity and those whose main activity was carried out with the aim of quitting tobacco (see Table 5).

Concerning the relationship between remaining in treatment for more than 3 months (Adherence3) and education level, the contrasts revealed differences in Adherence3 between participants with a high school and bachelor degree, with the high school group having the highest probability of remaining in treatment for more than 3 months. Likewise, statistically significant differences were found between participants with high school and those with a master/doctorate, with the high school group being the most likely to continue using the app. Finally, differences were also found between participants with a bachelor degree and those with a master/doctorate, with the master/doctorate group being the most likely to remain in treatment.

Regarding the relationship between staying in treatment for more than 3 months (Adherence3) and profession, the contrasts revealed differences in Adherence3 between public administration workers and unemployed participants, and between technical professionals and unemployed participants, with unemployed individuals having the highest probability of remaining in treatment (see Table 5).

Finally, regarding the relationship between the frequency of use of the Prescinde therapy app and education level, the contrasts showed differences in the frequency of app usage between participants with a bachelor degree and those with a master/doctorate, with the master/doctorate group having the highest probability of adherence (see Table 5).

## 4. Discussion

The increasing use of health applications on mobile phones has led to notable changes in user behavior, such as increased physical activity and the cessation of tobacco or cannabis use. These interventions provide users with unrestricted access to specialized information, overcoming geographic or programming limitations. Moreover, they offer advantages such as privacy, accessibility, and affordability, while effectively reaching a broader population, including younger individuals, women, and other users who are more reluctant to seek face-to-face help. This increased accessibility is particularly beneficial for users struggling with addictions to substances such as tobacco or cannabis, as they receive greater support in their efforts to quit. Identifying user-related factors, including sociodemographic characteristics, and examining their interaction with apps over time is a crucial challenge. Addressing this challenge would enable the development of targeted and cost-effective interventions with long-term sustainability for the health care system. The therapy app Prescinde shows promise as a potential solution in this regard.

This study aimed to analyze adherence to an intervention using a cell phone application as a function of sociodemographic variables (age, gender, educational level, and profession) as well as the main activity chosen by users (reduction in tobacco or cannabis consumption and increase in physical activity). We have analyzed adherence in terms of the duration of engagement and frequency of use of the Prescinde therapy app.

Regarding the adherence rate of the participants who used the app to stop using cannabis, 104 (62.65%) used the app during the first week, 60 (36.14%) during the three months, and 22 (13.25%) continued to use it after more than three months. These results, although apparently insignificant, are quite promising. It is well known in the tobacco/cannabis addiction literature that many people with these addictions do not even begin treatment, and if they do, they drop out prematurely [13,21]. These results indicate that the Prescinde therapy app could help to break down practical and emotional barriers in the treatment to quit tobacco use and increase physical exercise.

Regarding participants who used the app to quit cannabis use, our findings indicate that they predominantly engaged with the app during the first week and their likelihood of continued use declined until day 41. It is worth noting that the analysis of user engagement in cannabis cessation interventions through cell phone apps is relatively scarce. However, our results are promising and align with another study suggesting that a cannabis cessation app can enhance motivation to change [37]. However, this study only analyzed the percentage of users who downloaded the app. Similarly, for participants who used the app to quit tobacco use, we observed that their engagement persisted for approximately 141 days (20 weeks). These findings are consistent with another study that found a direct relationship between the commitment to app usage and tobacco abstinence [38], during which participants were connected between 4 and 26 weeks. Our results are also consistent with those obtained in face-to-face intervention programs [17], which demonstrated a 16.2% reduction in tobacco use and a 8.4% reduction in cannabis use over a 6-month follow-up period. Furthermore, when combining app therapy with face-to-face therapy, abstinence rates of approximately 65.7% were achieved within 7 days of treatment initiation [26]. It is worth noting that many smoking cessation apps available on various platforms show a low user engagement (16%) [43]. Consequently, the results of our study suggest that the Prescinde therapy app has had a positive impact on participants.

In contrast, it is worth noting that the participants who used the Prescinde therapy app to increase physical exercise demonstrated remarkable adherence, using the application for over three months, and reaching day 731. This finding is particularly significant because this study is the first to compare the use of a cell phone app for physical exercise with its use for smoking cessation (cannabis and/or tobacco). These results emphasize the significance of applying digital interventions for modifying sedentary behaviors and promoting physical activity. The results of the meta-analysis [34] indicate a consistent and positive association between engagement with digital health interventions for improved physical activity outcomes. Additionally, our findings shed light on the distinct profiles that emerge when aiming to modify habits and promote healthy behaviors such as increasing physical exercise or reducing harmful behaviors such as smoking tobacco or cannabis. Our study addresses concerns raised in the literature [5,14,18,24,27] regarding low adherence and high dropout rates in interventions aimed at quitting smoking tobacco and/or cannabis. Given the significant clinical and economic implications of cannabis and tobacco use, our study highlights the promising potential of the Prescinde therapy app as a cost-effective intervention with substantial health benefits. In particular, we have reported benefits in terms of reduced tobacco use at follow-up [10,18]. These findings suggest that the use of the therapy app Prescinde alone could be beneficial for tobacco cessation and increasing physical exercise.

Although we found no statistically significant differences in relation to gender, our analyses revealed interesting patterns regarding the likelihood of remaining engaged with the therapy app Prescinde. In the initial days of usage, men showed a lower probability of adherence. However, this pattern of results reversed from day 11, after which the probability of adherence was lowest for women. These findings are consistent with previous research suggesting that women face greater challenges in quitting smoking due to social pressures [27]. Moreover, other studies [25] have found that women have greater difficulties in achieving and maintaining abstinence from tobacco, which is potentially linked to higher levels of social anxiety and negative mood states compared to men [57]. However, it is worth noting that women can develop stronger therapeutic alliances than men [58]. Therefore, recognizing gender-specific factors is crucial in designing targeted smoking cessation interventions. Therefore, tailoring interventions based on gender-specific recommendations could enhance the effectiveness and accessibility of treatment and minimize barriers to treatment engagement.

When examining the likelihood of remaining in treatment for more than 3 months (Adherence3) as a function of age, our findings indicate that the oldest age group (26 to 62 years) showed a higher probability of long-term adherence than the youngest age group (17 to 18 years). These results are consistent with those reported by other authors [14,23,26]. In particular, a direct relationship was found between age and smoking cessation attempts [14], while smoking cessation has been shown to increase with age, ranging from 25.4% in individuals aged 16–34 years to 76.9% in those aged 65 years and older, based on a survey conducted in 2020 [23]. Moreover, the mean age of participants who quit smoking after 28 days of combined treatment (cell phone application and professional contact) was 41.07 years, with a range of 20 to 72 years. Another study found that the probability of quitting smoking increases from the age of 33 years onwards [26].

Regarding the relationship between long-term adherence (more than 3 months) and educational level, we found that the high school group had the highest probability of remaining in treatment compared to the master/doctorate group. Additionally, the master/doctorate group showed a higher likelihood of remaining in treatment than the bachelor degree group. These results are in line with previous research [23], which revealed that participants with primary education or lower showed lower rates of smoking cessation, while those educated to university level showed higher rates of smoking cessation (39.4%). Finally, regarding the relationship between the frequency of use of the Prescinde therapy app and education level, we found differences in the frequency of use of the application between participants with a bachelor degree and those with a master/doctorate, with the latter showing the highest probability of adherence. These results are consistent with other studies [26] showing that participants who quit smoking at 28 days with a combined treatment (cell phone application and contact with professionals) had studied at university.

Regarding the relationship between profession and remaining in treatment after more than 3 months, it was found that unemployed participants had a higher probability of adherence compared to public administration workers. Although these results are novel, they are in line with those found in the literature reviewed [28,29], highlighting the role of socioeconomic variables in smoking behavior.

However, this study has some essential limitations. First, there is a potential loss of information related to follow-up or post-treatment assessments. Our study only had a follow-up of 1 year for smoking and cannabis cessation activity and 700 days for physical exercise activity. Future research should consider using a longer follow-up period for interventions targeting the cessation of addictive behaviors. Second, we did not report cessation rates in the app, so the abstinence outcome is inferred based on the variables of adherence to treatment and frequency of use of the Prescinde therapy app. Third, the size of the group of participants whose main focus was stopping cannabis use was small, which makes drawing conclusions difficult. Therefore, future studies should increase the sample size of this specific group to support our results. Despite these limitations, our findings have several strengths and implications relevant to clinical practice. In particular, while the effectiveness of the therapy app Prescinde alone has been demonstrated, our findings highlight the importance of adopting more comprehensive and multicomponent approaches to treating cannabis and tobacco users.

## 5. Conclusions

An important aspect to consider is that our results offer valuable insights into the usage patterns of cell phone interventions over time. Specifically, these findings provide comprehensive information regarding the relationship between sociodemographic variables and utilization of the therapy app Prescinde for quitting smoking and increasing physical exercise.

Therefore, we can conclude that being male, being aged between 26 and 62, having a higher education, and being unemployed increase the probability of adherence after three months of using the Prescinde therapy app.

This study adds to the growing body of literature on the effectiveness of cell phone interventions in promoting behavioral changes related to quitting smoking tobacco and/or cannabis as well as increasing physical exercise. Importantly, our findings provide insights into the feasibility and impact of the Prescinde therapy app as a standalone intervention, focusing solely on the effects of the technology, without any professional or face-to-face intervention. In future studies, we will investigate, in addition to adherence to the app, the effectiveness of the mobile app in supporting smoking cessation.

## Figures and Tables

**Figure 1 healthcare-11-03121-f001:**
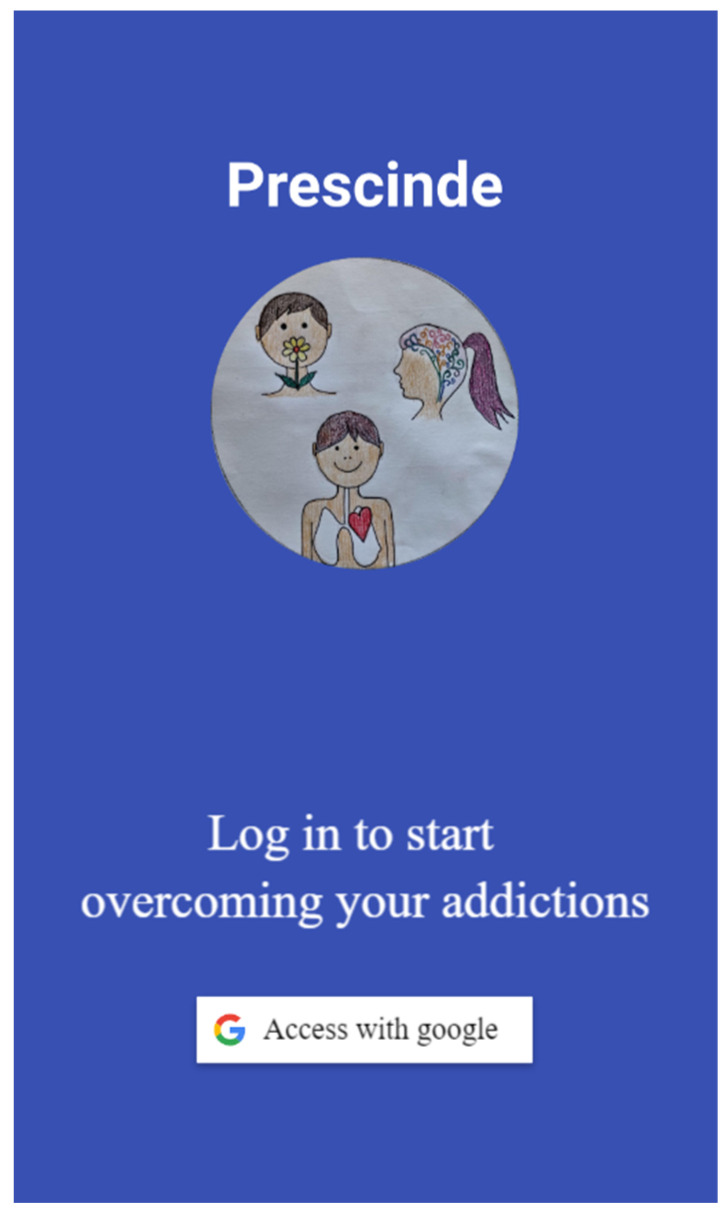
Initial screen of the therapy app Prescinde.

**Figure 2 healthcare-11-03121-f002:**
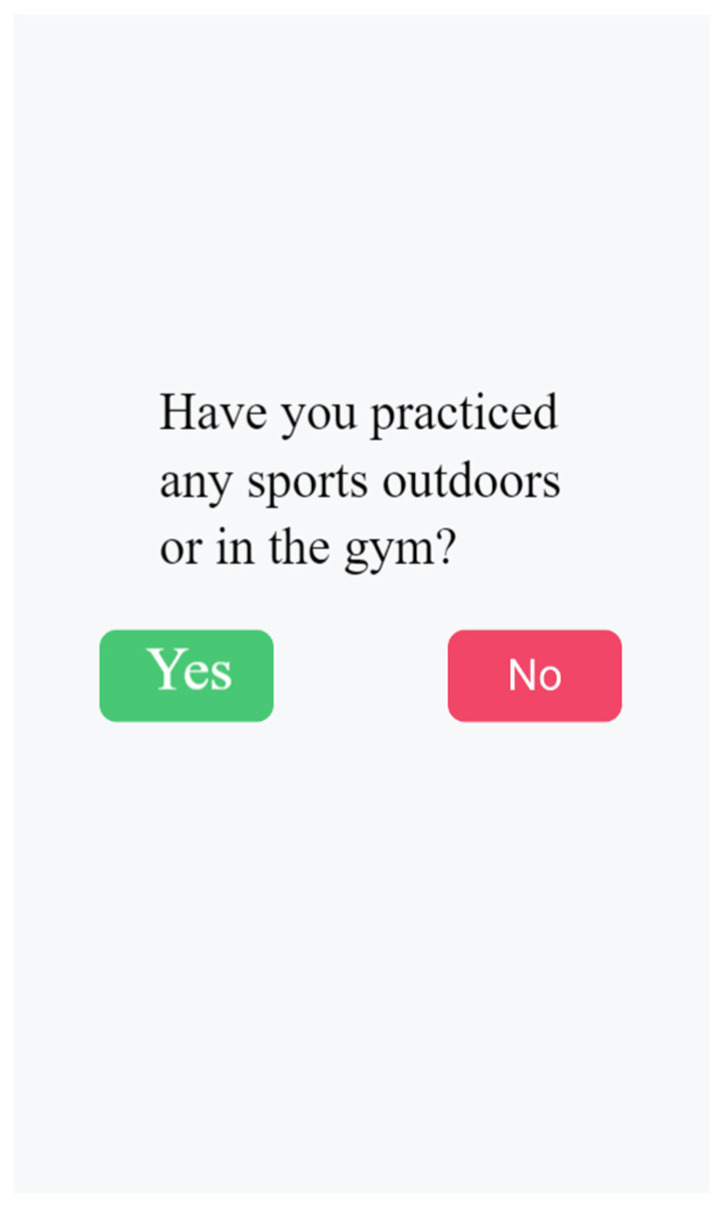
Question for the participant.

**Figure 3 healthcare-11-03121-f003:**
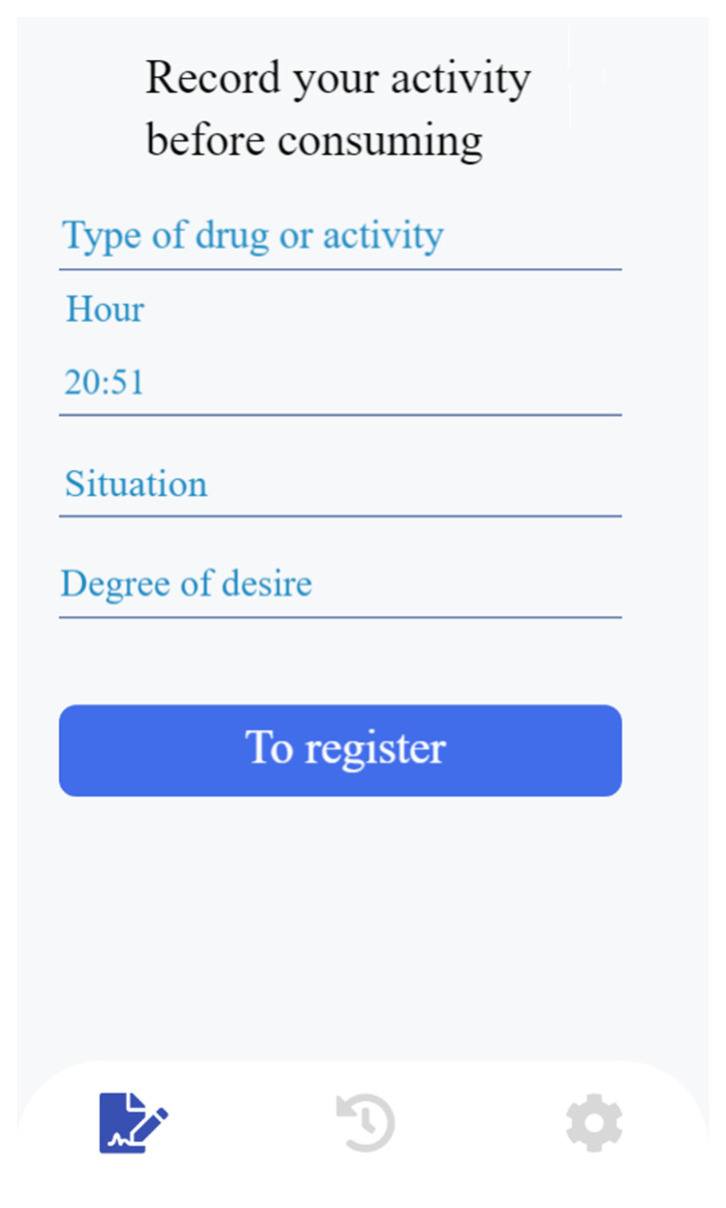
Recording consumption.

**Figure 4 healthcare-11-03121-f004:**
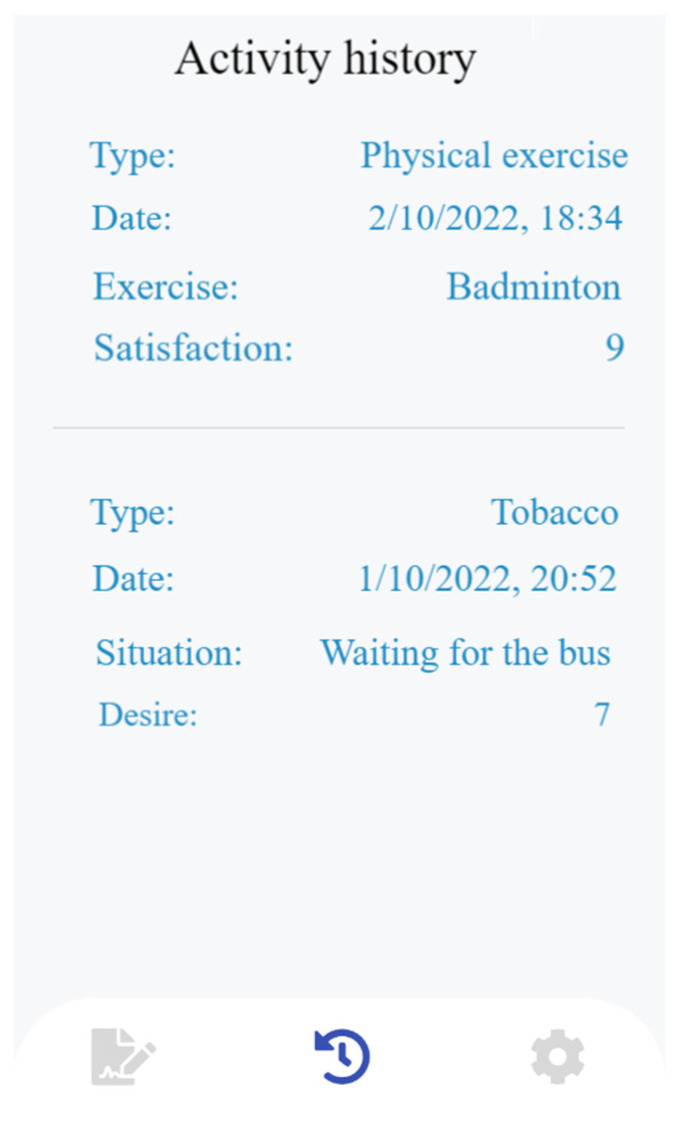
Activity history.

**Figure 5 healthcare-11-03121-f005:**
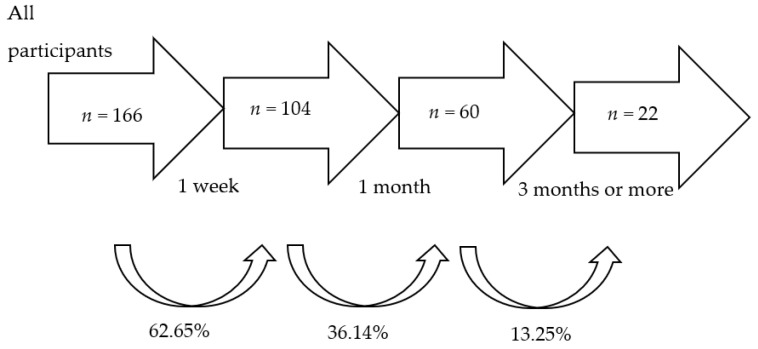
Participants in treatment over time, and adherence rates.

**Figure 6 healthcare-11-03121-f006:**
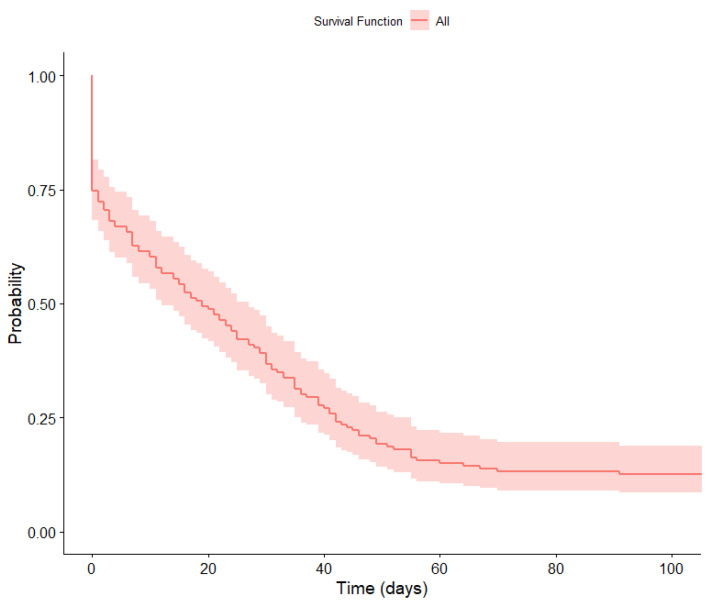
Survival function.

**Figure 7 healthcare-11-03121-f007:**
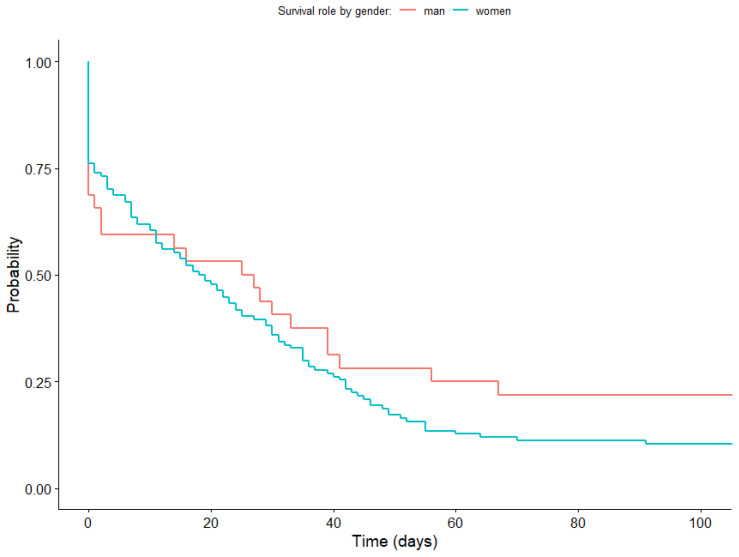
Survival function according to gender.

**Figure 8 healthcare-11-03121-f008:**
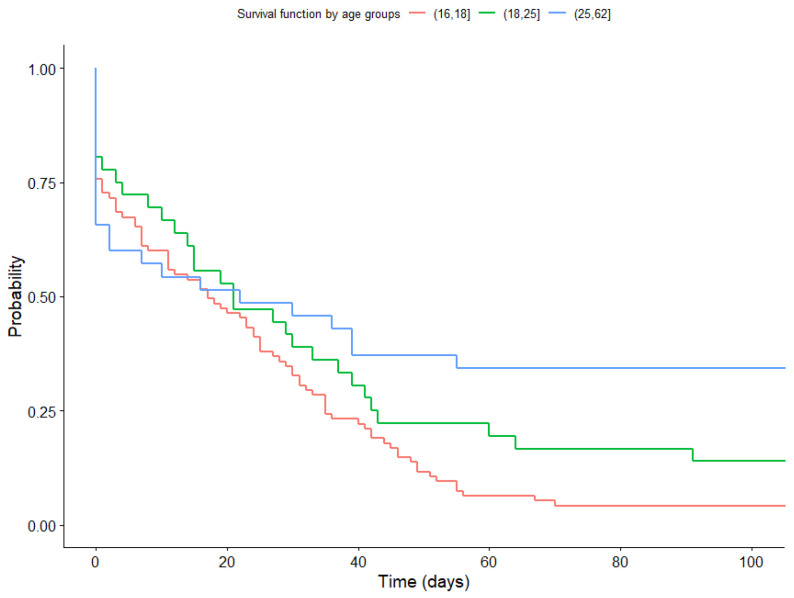
Survival function according to age group.

**Figure 9 healthcare-11-03121-f009:**
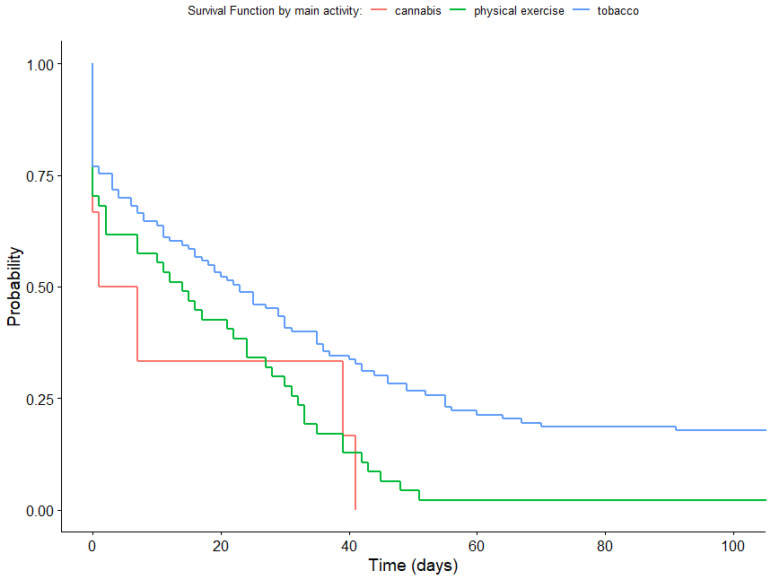
Survival function according to main use of the app (activity).

**Table 1 healthcare-11-03121-t001:** Sociodemographic characteristics, time using the Prescinde therapy app, and adherence levels 1 and 3.

	Gender		Age			Main Activity		
	Male	Female	(17–18)	(19–25)	(26–62)	Physical Exercise	Cannabis	Tobacco
Education level								
High School	12	70	58	16	8	62	4	16
Bachelor degree	13	49	37	10	15	38	1	23
Master/Doctorate	7	15		10	12	13	1	8
Currently studying								
No	9	13		4	18	11	3	8
Yes	23	121	95	32	17	102	3	39
Profession								
Public Administration	7	8		2	13	7		8
Agriculture/Fishing		1			1	1		
Trade		2		1	1	1		1
Private Company Manager	2			1	1		1	1
Hospitality/tourism	1	4	3		2	2	1	2
Technical professional	5	12	2	7	8	10	1	6
Transportation	1	1			2			2
Unemployed	16	106	90	25	7	92	3	27
Time								
1 week	13	49	37	10	15	38	4	20
1 month	6	37	27	12	4	29		14
3 months	6	33	27	8	4	25	2	12
More than 3 months	7	15	4	6	12	21		1
Adherence1 (1 month)								
Yes	13	48	31	14	16	46	2	13
No	19	86	64	22	19	67	4	34
Adherence3 (more than 3 months)								
Yes	7	15	4	6	12	21	0	1
No	25	119	91	30	23	92	6	46

**Table 2 healthcare-11-03121-t002:** Average frequency of use for participants who were active for a certain period of time.

	Gender		Age			Main Activity		
Time	Male	Female	(17–18)	(19–25)	(26–62)	Physical Exercise	Cannabis	Tobacco
1 week	4.385	2.612	2.405	1.600	5.333	2.263	5.500	3.850
1 month	14.333	9.621	9.519	10.833	13.750	8.897		13.143
3 months	96.333	22.091	20.889	44.375	97.000	15.640	25.000	72.166
More than 3 months	153.000	118.067	118.750	132.167	131.167	134.476		18.000

Note. Frequency of use is the variable used to refer to the number of times a user has logged on to the therapy app Prescinde.

**Table 3 healthcare-11-03121-t003:** *p*-value of the Mann–Whitney–Wilcoxon test and Kruskal–Wallis test.

	Frequency of Use	Time
Gender	0.133	0.726
Studying	0.665	0.297
Working	0.828	0.245
Tobacco Consumption	0.015 *	0.337
Cannabis Consumption	0.012 *	0.197
Age	0.048 *	0.416
Main activity	0.856	0.059
Education level	0.017 *	0.021 *
Profession	0.196	0.216

Note: Mann–Whitney–Wilcoxon test for variables with two categories and Kruskal–Wallis test for variables with more than two categories. * *p*-value < 0.05.

**Table 4 healthcare-11-03121-t004:** *p*-value of the Chi-square test.

	Adherence1	Adherence3
Gender	0.764	0.190
Studying	1.000	0.081
Working	0.713	0.160
Tobacco Consumption	0.338	0.010 **
Cannabis Consumption	0.388	0.010 *
Age	0.373	0.010 **
Main activity	0.292	0.012 *
Education level	0.148	0.010 **
Profession	0.648	0.034 *

Note: The categories of the variables are shown in Table 1. * *p*-value < 0.05, ** *p*-value < 0.01.

**Table 5 healthcare-11-03121-t005:** *p*-value of the Mann–Whitney–Wilcoxon post hoc test and Chi-square test.

Variables		Comparative Groups	*p*-Value
Age	Adherence3	(17–18)–(19–25)	0.043 *
		(17–18)–(26–62)	0.010 **
		(19–25)–(26–62)	0.152
Main activity	Adherence3	Cannabis–Physical exercise	0.539
		Cannabis–Tobacco	0.010 **
		Physical exercise–Tobacco	0.012 *
Education level	Frequency of use	High School–Bachelor degree	0.112
		High School–Master/Doctorate	0.466
		Bachelor degree–Master/Doctorate	0.039 *
Education level	Adherence3	High School–Bachelor degree	0.045 *
		High School–Master/Doctorate	0.046 *
		Bachelor degree–Master/Doctorate	0.010 **
Profession	Adherence3	Public administration–Unemployed	0.012
		Technical Professional–Unemployed	0.026 *

Note. Mann–Whitney–Wilcoxon post hoc test for quantitative variables and Chi-Square test for qualitative variables. * *p*-value < 0.05, ** *p*-value < 0.01.

## Data Availability

Data available on request due to restrictions of privacy. The data presented in this study are available on request from the corresponding author. The data are not publicly available due to their containing information that could compromise the privacy of research participants.

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
