# Peer review of "Use of the Therapy App Prescinde for Increasing Adherence to Smoking Cessation Treatment"

_healthcare, 2023, doi:10.3390/healthcare11243121_

Round 1

Reviewer 1 Report

Comments and Suggestions for Authors

The paper by Lopez-Torrecillas and colleagues from the University of Granada, Spain describes adherence to a mobile phone application (App-therapy Prescind) as a function of a number of social demographic variables and according to the primary activities supported by the app; namely, reducing tobacco or cannabis use or increasing physical exercise.

There is a growing interest in the use of phone apps for smoking cessation as well as other addictions and this paper adds to the body of information about the design and adherence to the app according to population demographics.

The application is aimed at individuals wishing to quit smoking or cannabis use or increase exercise activity and its content includes three components: self-report, motivational phrases and goal setting.

The population for the study was recruited through webpages of the of the Occupational Risk Prevention Service and the Psychology clinic of the University of Granada during the Covid-19 confinement period.  The population is skewed towards women (80%) and younger individuals with 57% between 17 and 18 years of age. Furthermore, the main use of the app was for physical exercise (60%) whereas quitting smoking cannabis represented only 3.6% of the participants. The main objective of the study was to determine whether users continued to use the application for the main purpose they identified (reducing tobacco or cannabis consumption; increasing physical exercise). The frequency of app use was analysed according to sociodemographic variables and adherence levels were assessed at 1 and 3 months.

Although there is a need for more information about mobile phone apps and their utility in addressing smoking cessation, the study has a number of weaknesses. First of all the sample size is small (n=166) relative to the large number of variables that the authors wish to explore and made more complicated by looking at smoking tobacco, cannabis and exercise. It also makes the paper excessively long and difficult to read. This reviewer would suggest that the manuscript be broken down into several manuscripts. The information would be of particular value for smoking cessation. The number of individuals using cannabis is extremely small and I do not agree with the author’s statement that cannabis poses a significant health risk and is a major contributor to causes of death worldwide. There is no evidence at this time that cannabis is responsible for any cancers or other chronic illnesses. This may become evident I  the future but we cannot say so now.

The method of recruitment of participants to the study may well introduce bias particularly when individuals were recruited from a Psychology clinic rather than the generap population.  It would be very difficult to generalize form this study.

The authors describe the motivating phrases used if the user chose to quit smoking or increase physical exercise but for cannabis it does not list any phrases but only some of the negative effects of cannabis consumption. In a revised publication, a table showing the phrases incorporated into the app would add greater clarity. In describing the third component of the app which focuses on goals related to seven areas, there are an enormous number of questions. Are users of the app only intended to read them to provide motivation or do they actually have to answer the questions? Either way it seems like an enormous burden. Perhaps this explains why 21% participants only used the app once.

A major weakness of the study is that cessation rates are not documented but only inferred based on the usage of the app.

Author Response

Dear reviewer,

thank you very much for taking the time to review this manuscript. Please find the detailed responses in the attachment files.

Reviewer 2 Report

Comments and Suggestions for Authors

Overall, the paper is well-written. I have some minor suggestions. 

1. Page 2 of 21

Line number: 59-60

"The most common method for smoking cessation is without assistance."

Authors may need to provide examples of smoking cessation without assistance.

2. Page 2 of 21

Line number: 62

"Treatment for quitting cannabis and/or tobacco use continues to pose challenges."

Authors may need to specify the type of treatment, such as nicotine replacement therapy or nicotine patch.

3. Page 3 of 21

Line number: 103-104 

"However, a content analysis on smoking cessation apps has revealed several shortcomings"

Authors may need to identify or list certain shortcomings or limitations associated with the discussed content analysis on smoking cessation apps.

4. Page 4 of 21

Figure 1 , 2, 3, and 4

Authors may need to include English translations of the content presented in the figures1-4 .

5. Page 18 of 21

The current study did not utilize face-to-face intervention. One potential future direction could involve comparing the effectiveness of App-therapy Prescinde with in-person, face-to-face intervention. The authors may want to consider incorporating this potential direction into either the Discussion section or the Conclusion section of the paper.

Author Response

Thank you very much for taking the time to review this manuscript. Please find the detailed responses in the attached file.

Round 2

Reviewer 1 Report

Comments and Suggestions for Authors

The authors acknowledge that the sample size of the study was small (n = 166) but indicate that this was due to recruitment issues during the Covid pandemic. Nonetheless this does not change the fact that the study is small with numerous analyses of sub- groups which makes drawing conclusions difficult or impossible.

The manuscript actually states that cannabis poses a significant health risk and is a major contributor to causes of death worldwide. The response is that tobacco and cannabis are used to gather. Although this happens, cannabis is also used alone and in literature reviews has not been associated with the development of cancer or is it identified as increasing deaths. The authors provide information about the negative impacts of cannabis on mental health, but these do not support their assertion that cannabis is a major contributor to causes of death worldwide. Rewording of the text is still required

The response to the reviewer’s concern about the method of recruitment and the risk of bias should be incorporated into the text of the manuscript in the method section.

As the number of cannabis users was extremely small, the authors have eliminated the discussion of cannabis in the conclusions except to note that it is a limitation in the manuscript.

The authors should acknowledge that the inability to report cessation rates in their study is a weakness of the study and indicate in the discussion or conclusions that future studies will investigate the effectiveness of the mobile app in supporting smoking cessation.

Author Response

Dear reviewer,

Authors have made the modifications following his suggestions in order to a good presentation of the paper.

Please find enclosed herewith a revised version of the article with the changes highlighted to provide help to catch the improvements.

It is hope that the present version will be finally acceptable to you.

With kind regards,

Authors
